# Emerging Roles of RNA-Binding Proteins in Inner Ear Hair Cell Development and Regeneration

**DOI:** 10.3390/ijms232012393

**Published:** 2022-10-16

**Authors:** De-Li Shi, Xiao-Ning Cheng, Audrey Saquet, Raphaëlle Grifone

**Affiliations:** 1Department of Medical Research, Affiliated Hospital of Guangdong Medical University, Zhanjiang 524001, China; 2Laboratory of Developmental Biology, CNRS-UMR7622, Institut de Biologie Paris-Seine (IBPS), Sorbonne University, 75005 Paris, France; 3Institute of Clinical Medicine, Central People’s Hospital of Zhanjiang, Guangdong Medical University, Zhanjiang 524000, China

**Keywords:** inner ear, cochlear hair cells, RNA-binding proteins, post-transcriptional regulation, sensorineural hearing loss, zebrafish neuromasts, regeneration

## Abstract

RNA-binding proteins (RBPs) regulate gene expression at the post-transcriptional level. They play major roles in the tissue- and stage-specific expression of protein isoforms as well as in the maintenance of protein homeostasis. The inner ear is a bi-functional organ, with the cochlea and the vestibular system required for hearing and for maintaining balance, respectively. It is relatively well documented that transcription factors and signaling pathways are critically involved in the formation of inner ear structures and in the development of hair cells. Accumulating evidence highlights emerging functions of RBPs in the post-transcriptional regulation of inner ear development and hair cell function. Importantly, mutations of splicing factors of the RBP family and defective alternative splicing, which result in inappropriate expression of protein isoforms, lead to deafness in both animal models and humans. Because RBPs are critical regulators of cell proliferation and differentiation, they present the potential to promote hair cell regeneration following noise- or ototoxin-induced damage through mitotic and non-mitotic mechanisms. Therefore, deciphering RBP-regulated events during inner ear development and hair cell regeneration can help define therapeutic strategies for treatment of hearing loss. In this review, we outline our evolving understanding of the implications of RBPs in hair cell formation and hearing disease with the aim of promoting future research in this field.

## 1. Introduction

The primordium of the inner ear, called the otic placode, arises on either side of the hindbrain region as a thickening of the ectoderm. The otic placode undergoes invagination to change into an otic cup that subsequently closes to form a transient structure, the otic vesicle or otocyst [1]. The mediolateral, anteroposterior, and dorsoventral axes of the inner ear, which are sequentially specified during otocyst formation, are important for the development of different sensory structures [2,3]. The mammalian inner ear is formed by a snail-shaped cochlea, which is essential for hearing, and by a vestibular system necessary for maintaining balance. Strikingly, both the cochlear and the vestibular epithelia display highly polarized hair cells with uniformly oriented stereociliary bundles on their apical surfaces, which is critical for auditory and vestibular mechanosensation [4]. For example, the organ of Corti in the cochlea consists of one row of inner hair cells and three rows of outer hair cells, which are separated by specialized supporting cells. Defective morphogenesis and damage or loss of hair cells causes sensorineural hearing loss (SNHL) in humans [5], one of the most common sensory hearing disabilities in the world [6]. At present, transcriptional regulatory mechanisms underlying the development of the auditory system has been relatively well documented [7]. Several transcription factors including Sox2, Otx2, Atoh1, and Pou4f3, as well as different signaling pathways, such as Wnt, Shh, Notch, BMP4, and FGF, are critically involved in the formation of the otocyst [1,8]. They function sequentially in the specification of progenitors, the proliferation of prosensory cells, and the differentiation of hair cells, thus making an important contribution to the development of inner ear structures and cell-type differences. The same transcription factors and signaling pathways function to modulate sensory hair cell regeneration in non-mammalian vertebrates [9]. However, the post-transcriptional control of inner ear development as well as of hair cell function and regeneration remains elusive, mostly owing to limited studies of the functional genes involved in this regulatory process.

RNA-binding proteins (RBPs) are involved in the post-transcriptional regulation of gene expression and function. Through interactions with their protein partners and target transcripts, including both coding and non-coding RNAs, they control RNA metabolism at multiple levels, from pre-mRNA splicing to mRNA transport, localization, stability or degradation, polyadenylation, and translation [10]. Therefore, RBPs contribute to the diversity of the proteome, the stage- and tissue-specific expression of protein isoforms, and the maintenance of protein homeostasis within the cell. Dysfunctions of RBPs have been linked to various human diseases [10,11]. In recent years, accumulating evidence has revealed emerging roles of post-transcriptional regulatory processes in coordinating inner ear development and auditory function. Several RBPs are critically required for pre-mRNA splicing and mRNA stability in hair cells. In particular, LIN28A-B, RBM24, SFSWAP, SRRM4, and ESRP1 play key roles in hair cell differentiation and regeneration. Functional analyses using different animal models and identification of genetic mutations in humans suggest that loss of their activity affects inner ear development and leads to hearing impairment. For example, mutations of the *ESRP1* gene in humans disrupts inner ear-specific alternative splicing events and segregates with SNHL (OMIM#618013), suggesting a critical role of this RBP in auditory function [12]. This review summarizes RBP-mediated post-transcriptional regulation of hair cell development and regeneration. As an outcome, it helps identify gaps in our understanding of this field and proposes directions for future investigation.

## 2. RBP-Mediated Post-Transcriptional Regulation in Inner Ear Hair Cell Development and Regeneration

RBPs control gene expression at the RNA level through different mechanisms (Figure 1). Significant advances have been made in understanding the post-transcriptional regulation of hair cell differentiation, regeneration, and function. Several RBPs that play important roles in these processes have been identified through functional analyses using animal models and by identification of hearing loss genes in humans (Table 1). The following sections detail their implications in hair cell development and hearing disease.

### 2.1. LIN28 in Cochlear Hair Cell Development and Regeneration

LIN28 (LIN28A and LIN28B) proteins are highly conserved small cytoplasmic RBPs that function as pluripotency factors, regulating the transition from self-renewal to a differentiated cell fate [40]. Consistent with this activity, functional analyses in mice suggest that Lin28B plays an important role in hair cell development and regeneration. In the cochlea of mouse embryos, it is highly expressed in prosensory cells and down-regulated at the onset of hair cell differentiation. Prolonged expression of Lin28B delays prosensory cell cycle exit and prevents hair cell differentiation, suggesting that it functions to increase hair cell production [14]. Interestingly, Lin28B inhibits the processing of mature *let7* miRNA, which functions to induce cell cycle exit in progenitor cells [14]. Therefore, the antagonistic actions of Lin28B and *let7* miRNA coordinate the timing of prosensory cell cycle withdrawal for hair cell differentiation. In neonatal murine cochlear organoids and explants, Lin28B antagonizes the activity of *let7* miRNA and increases Akt-mTORC1 signaling to promote hair cell regeneration from immature supporting cells by inducing their de-differentiation and proliferation as well as by directly converting them into hair cells [15]. Thus, Lin28B functions in hair cell regeneration through mitotic and non-mitotic mechanisms, which are dependent on mitotic division or trans-differentiation of supporting cells into hair cells, respectively. The precise mechanism by which Lin28B and *let7* miRNA regulate mTORC1 activity in cochlear epithelial cells awaits further investigation. It is possible that Lin28B directly promotes mRNA translation of mTOR pathway genes or relieves *let7*-mediated repression of their translation [41]. In addition, Lin28B functions to enhance the regenerative competence of maturing supporting cells in the cochlea through cooperation with Follistatin, which inhibits Lin28B-induced TGF-ß signaling that can trigger proliferative quiescence [16]. This suggests that coactivation of Lin28B and Follistatin may represent an endogenous mechanism mediating reprogramming of supporting cells for hair cell regeneration.

Lin28A is required for hair cell regeneration in the mammalian cochlea, and may function in redundant processes with Lin28B [15,16]. Studies using a zebrafish model further illustrate an important role of Lin28A in the recovery of progenitor cells. It has been shown that severe injury with loss of both progenitors and hair cells induces robust transient upregulation of Lin28ab (a zebrafish ortholog of human LIN28A) in regenerating neuromasts through activation of Yap, which directly binds to the *lin28ab* promoter to initiate its transcription in hair cell precursors [13]. Furthermore, Lin28ab inhibits the function of *let7* miRNA to activate the Wnt/ß-catenin pathway for progenitor proliferation and hair cell regeneration [13]. Therefore, studies using different vertebrate models have demonstrated a role for Lin28 paralogs in promoting prosensory cell proliferation and in initiating hair cell regeneration. However, it is unclear how they post-transcriptionally regulate target gene expression in prosensory cells and supporting cells during hair cell development and regeneration. Lin28 and *let-7* miRNA are mutually antagonistic, repressing the expression of each other, and they function as a regulatory pair in stem cell plasticity, somatic cell reprogramming, and tissue regeneration [40]. Thus, further investigation is necessary in order to better understand how Lin28 and *let-7* miRNA interact to modulate gene expression for hair cell regeneration.

### 2.2. RBM24 Regulates mRNA Stability and Pre-mRNA Splicing in Hair Cells

RNA binding motif protein 24 (RBM24) contains a highly conserved RRM (RNA recognition motif) at the N-terminus that binds to GU-rich sequences in target transcripts [42]. It displays restricted tissue-specific expression patterns in all vertebrate species [43]. In addition to striated muscles, Rbm24 is strongly expressed in head sensory organs, including the otic vesicle, lens, and olfactory vesicle [44,45]. In the inner ear of neonatal mice, Rbm24 expression is detected in a subset of hair cells and is directly regulated by the transcription factor Atoh1 [46]. This suggests that Rbm24 may be a transcriptional target and function downstream of Atoh1 in hair cell differentiation. Immunofluorescence staining shows that Rbm24 protein highly accumulates in the cytoplasm, with weak localization in the nucleus of inner ear hair cells (Figure 2). It co-localizes with Myo7A in mechanosensory cells of the auditory and vestibular systems, suggesting that it may play a role in sensory hair cell differentiation and function [44]. 

The zebrafish genome harbors *rbm24a* and *rbm24b* paralogs, although only *rbm24a* is expressed in the anterior and posterior maculae as well as in the posterior lateral line primordium that migrates caudally to periodically deposit neuromasts [17]. Single cell RNA-seq analysis indicates that the expression of zebrafish *rbm24a* dynamically changes from non-cycling progenitors to differentiated hair cells; thus, it may participate in the differentiation program [48]. We have made the first demonstration of a functional role of Rbm24a in inner ear development by analyzing the zebrafish *rbm24a* mutant line generated using the CRISPR/Cas9 approach [17,45]. Importantly, loss of Rbm24a delays inner ear development and impairs hair cell morphogenesis in both the inner ear and in the lateral line neuromasts (Figure 3), resulting in hearing and balancing deficits [17]. These defects are correlated with a reduced expression level of several zebrafish orthologs of human deafness genes, including *smpx* (small muscle protein X-linked), *gsdmeb* (gasdermin Eb), and *otofa* (otoferlin a). Further analysis suggests that Rbm24a deficiency leads to reduced stability of the corresponding mRNAs [17,18]. Thus, *smpx*, *gsdmeb*, and *otofa* are possible target mRNAs of Rbm24a for hair cell morphogenesis and function. Consistently, zebrafish *smpx* morphants show structural alterations of stereocilia and kinocilia in hair cells as well as impaired hair cell activity [49].

Consistent with a multifaceted post-transcriptional regulator, RBM24 plays an important role in alternative splicing. It has been shown that deletion of Rbm24 in mice affects the inclusion of the inner ear-specific exon 68 in *Cadherin 23* (*Cdh23*) mRNA, leading to hearing loss and defective motor coordination [19,20]. Cdh23 and Cdh15 are important components of the tip links that interconnect the mechanosensory stereocilia and the kinocilium in the hair bundle for mechanotransduction [50]. Mutations of the *CDH23* gene are responsible for human Usher syndrome 1D (OMIM#601067) and non-syndromic autosomal recessive deafness DFNB12 (OMIM#601386) [51,52,53,54]. Because the inner ear-specific exon 68 of *CDH23* gene encodes amino acids involved in interaction with other proteins in hair cells [55], dysfunction of RBM24 may cause a chain of events that impairs hair cell development and function. In mice, it has been shown that RBM24 promotes muscle-specific alternative splicing by preventing the suppression of exon inclusion mediated by splicing repressors PTBP1 (polypyrimidine tract-binding protein 1), also known as hnRNP I (heterogeneous nuclear ribonucleoprotein I), and hnRNP A1/A2 [56]. Rbm24 likely regulates the inclusion of exon 68 in *Cdh23* mRNA through a similar mechanism because PTBP1 seems to repress inclusion of this exon [19]. Further supporting its functional importance in inner ear development, conditional knockout of Rbm24 in mice has been shown to affect the survival of outer hair cells in the cochlea [21]. Therefore, Rbm24 plays critical roles in the post-transcriptional regulation of hair cell morphogenesis and function. However, although Rbm24 protein is predominantly localized to the cytoplasm of differentiated hair cells in neonatal mice, it is unclear how it is distributed in prenatal sensory hair cells or how it regulates post-transcriptional events during the early stages of inner ear development. Indeed, Rbm24 is expressed in the otic vesicle during early development, at least in E10.5 mouse embryos [43]. Dynamic subcellular trafficking and post-transcriptional regulatory functions of Rbm24 have been demonstrated during muscle cell differentiation and regeneration in mice [57]. Thus, it is of interest to examine whether a similar situation exists during the process of hair cell differentiation in the early embryo. It is worth understanding how Rbm24 acts downstream of transcription factors, such as Atoh1, to relay their activity for hair cell development and regeneration.

### 2.3. SFSWAP Functions in Growth and Patterning of Inner Ear Sensory Organs

The splicing factor SWAP (SFSWAP) is a mammalian homolog of *Drosophila* suppressor-of-white-apricot that displays RNA-binding activity and regulates alternative splicing of *CD45* and *Fibronectin* pre-mRNAs in COS cells [58]. In mice, Sfswap is widely expressed in the developing inner ear, then becomes more restricted in the cochlea and the spiral ganglion at birth [22]. Loss of Sfswap function leads to defective inner ear patterning, resulting in reduced numbers of outer hair cells and ectopic inner hair cells in the cochlea as well as smaller cristae and maculae in the vestibular system. This suggests that Sfswap plays an important role in the accurate formation of sensory organs and proper patterning of mechanosensory hair cells. Accordingly, homozygous *Sfswap* mutant mice show mild hearing loss, changed vibratory responses of the organ of Corti, and circling behavior [22,23]. Consistent with its involvement in the expression of Notch pathway genes in the inner ear, *Sfswap* genetically interacts with *Jagged1* in cochlear patterning [22]. However, it is unclear how Sfswap regulates inner ear-specific alternative splicing or whether it is involved in other aspects of the RNA metabolism. Thus, identification of its target RNAs should provide insights into the post-transcriptional mechanisms by which it functions in cochlear patterning. In particular, it is worth examining how it is involved in the post-transcriptional regulation of Notch pathway genes.

### 2.4. “Noise/Damage-Related” RBPs in Hearing Loss

There is evidence that RBPs are potentially implicated in noise/damage-induced hearing loss. Quaking (QKI or QK) proteins contain a STAR (signal transduction and activation of RNA) domain and bind to ACUAA motifs in the 3′-UTR to regulate mRNA function [59,60]. The *QKI* gene produces three major protein isoforms through alternative splicing, namely, QKI-5, QKI-6, and QKI-7. In the cochlea of postnatal and adult mice, Qki-6 and Qki-7 isoforms are mostly accumulated in the cytoplasm of glial cells surrounding spiral ganglion neurons and auditory nerve fibers. It has been shown that auditory nerve degeneration and hearing deficiency in mice caused by noise exposure are associated with a decreased expression of Qki proteins and their numerous possible target genes [24]. Conditional knockout of the *Qki* gene to prevent the expression of all Qki isoforms in cochlear glial cells leads to hearing deficiency due to defective myelination of spiral ganglion neurons and auditory nerve fibers, suggesting that dysfunction of Qki-mediated regulatory process may be an important early target of the noise response [24].

CAPRIN1 (cytoplasmic activation/proliferation-associated protein 1) is a ubiquitously expressed RBP originally identified in lymphocytes and hematopoietic progenitor cells [61]. It contains an arginine-glycine-glycine (RGG) box that functions as an RNA-binding motif for G-rich RNA targets. In the cochlea of postnatal rats, Caprin1 is detected in both hair cells and supporting cells, and it may act as a transcriptional target of Pou4f3 to regulate hair cell survival in response to ototoxic damage [25]. Pou4f3 normally represses the expression of the *Caprin1* gene by directly binding to its regulatory sequence. It has been proposed that ototoxin-induced reduction of Pou4f3 expression in cochlear hair cells could lead to upregulation of the Caprin1 protein, which can form stress granules and may sequester mRNAs of pro-apoptotic genes to inhibit their translation, thereby preventing ototoxin-induced death of sensory hair cells [25,26]. Moreover, inner ear-specific deletion of Caprin1 in mice affects synapse formation between inner hair cells and spiral ganglion neurons, leading to early onset progressive hearing loss. Loss of Caprin1 impairs the ability of the cochlea to recover from noise exposure, suggesting that it is required for the maintenance of the auditory function [27]. Therefore, QKI and CAPRIN1 may be potential therapeutic targets for noise- and ototoxin-induced hearing loss.

### 2.5. Post-Transcriptional Inactivation of REST Transcriptional Repressor by SRRM4-Regulated Exon Inclusion

SRRM4 (serine/arginine repetitive matrix 4), also known as nSR100 (neural-specific SR-related protein of 100 kDa), belongs to a large family of proteins harboring serine/arginine (SR)-repeats. It displays RNA-binding activity and regulates a network of brain-specific exons in genes with important function for neural cell differentiation [62]. Bronx waltzer (*bv*) mice show degeneration of cochlear inner hair cells, defective synaptogenesis, deafness, and impaired balance [63,64,65]. Positional cloning maps the *bv* mutation to the *Srrm4*/*nSR100* locus, and transcriptomic analysis indicates that loss of Srrm4 specifically disrupts alternative splicing events in the inner ear [28]. Knockdown of Srrm4 in zebrafish leads to hair cell degeneration in the neuromasts [28]. These observations suggest that Srrm4-regulated alternative splicing plays a conserved role in hair cell development and hearing function.

REST (restrictive element-1 silencing transcription factor), also known as NRSF (neuron-restrictive silencing factor), is a transcriptional repressor that silences the expression of a large number of neural genes in non-neural tissues. However, its function is specifically inactivated in neurons and inner ear hair cells, thereby allowing the expression of neural genes in both cell types [66]. In mouse mechanosensory hair cells, regulated inclusion of the frameshift-causing exon 4 into the *Rest* mRNA is essential for its inactivation, while skipping of this exon allows the expression of a functional Rest protein, causing hair cell degeneration and deafness [29]. Srrm3 and Srrm4 are differentially expressed in inner ear hair cells, and regulate the splicing-dependent inactivation of Rest protein [30]. They likely promote exon incorporation in the *Rest* mRNA in neural cells by directly preventing PTBP1-mediated repression of exon inclusion [67]. In the utricle, Srrm3 expression is dependent on Srrm4-mediated inactivation of Rest function, whereas in cochlear outer hair cells it is independent of Srrm4 due to a transient down-regulation of Rest activity in these cells, as the *Srrm3* gene is itself a target of Rest-mediated transcriptional repression [30]. Therefore, Srrm4 is required for inactivating Rest protein in cochlear inner hair cells and vestibular hair cells, while Srrm3 and Srrm4 function redundantly to inhibit Rest activity in cochlear outer hair cells. This may explain the normal morphology of outer hair cells in the *bv* mutant with loss of Srrm4 function [28,30]. Consistently, combined loss of Srrm3 and Srrm4 in mice causes complete loss of hair cells in the inner ear. Importantly, the DFNA27 mutation (OMIM#612431) that changes the splicing acceptor site of exon 4 in the human *REST* gene prevents SRRM4-dependent alternative splicing of this exon and causes progressive hearing loss [29], further illustrating the requirement of SRRM4-mediated post-transcriptional regulation in REST inactivation for hair cell development and function.

### 2.6. Mutations of the ESRP1 Gene in Humans Cause Alternative Splicing Defects and Hearing Loss

ESRP1 (epithelial splicing-regulatory protein 1), previously known as RBM35A, belongs to the hnRNP family of RBPs and regulates alternative splicing of *FGFR2* pre-mRNA in epithelial tissues [68]. Importantly, frameshift and missense mutations of the *ESRP1* gene have been identified in individuals with profound bilateral SNHL by whole-exome sequencing [12]. Analysis of ESRP1-dependent alternative splicing events using patient-derived iPSCs (induced pluripotent stem cells) indicates altered exon inclusions for several mRNAs, including *ENAH*, *NF2, RALGPS2*, and *ARHGEF11* [12], suggesting that mutations of *ESRP1* cause hearing loss by disrupting alternative splicing. Loss of Esrp1 in mice causes defective inner ear morphogenesis and prevents cochlear hair cell differentiation, mostly by disrupting epithelial-specific splicing of *Fgfr2* pre-mRNA in cochlear epithelial cells, leading to inappropriate expression of the mesenchymal Fgfr2-IIIc isoform that displays a different ligand-binding specificity [12]. These data highlight the importance of ESRP1-mediated alternative splicing in inner ear development and link *ESRP1* mutations with SNHL (OMIM#618013). 

There are two highly homologous *Esrp* genes in vertebrates, *Esrp1* and *Esrp2* [68]. Although no mutations of *ESRP2* have been associated with SNHL, there is evidence that Esrp1 and Esrp2 display both distinct and redundant functions in regulating the epithelial splicing program during tissue and organ morphogenesis in mice [69]. Moreover, at least in zebrafish and chicks, Esrp2 is expressed in the otic placode and along the tonotopic axis [70,71]; thus, there is a possibility that it may have a function in inner ear development.

### 2.7. Possible Role of Musashi1 in the Maintenance of Stem Cell Fate for Hair Cell Regeneration

Musashi1 (Msi1) is a neural RBP with two RRMs and is expressed in neural stem/progenitor cells [72]. In mice, expression of the Msi1 protein is present in all otocyst cells between E12 and E14, and is absent in vestibular hair cells at E14 and in cochlear hair cells at E16 [31]. However, Msi1 can be detected in different supporting cells during postnatal and adult life, including pillar and Deiters cells [31,32,33]. Interestingly, the subcellular localization of Msi1 in supporting cells undergoes cytoplasm to nucleus translocation during the first two weeks after birth, implying a dynamic function in inner ear development [31]. In the utricle of chick embryo, Msi1 is localized to the basal side of supporting cells and is upregulated along with downstream genes of the Notch pathway following aminoglycoside-induced ototoxic damage [34]. During the spontaneous regeneration process of vestibular hair cells in guinea pigs, Msi1 becomes redistributed in the cytoplasm of supporting cells, which undergo asymmetric division to produce hair cells [35]. These observations suggest that nuclear Msi1 may play a role in the maintenance of stem cell fate. However, functional analyses are needed to determine the role of Msi1 in hair cell regeneration. Because Msi1 contributes to self-renewal of neural stem cells by increasing Notch signaling activity through translational repression of *Numb* mRNA [72], it is of interest to examine whether this post-transcriptional mechanism functions to maintain stem cell fate in the inner ear

### 2.8. Other RBPs Implicated in Inner Ear Development and Hair Cell Regeneration

Zebrafish possess sensory hair cells in both the ear and in lateral line neuromasts (see Figure 3). They are structurally and functionally equivalent to hair cells in the mammalian inner ear, and are subjected to similar molecular regulation [73]. In contrast to mammals, zebrafish can robustly regenerate inner ear and neuromast hair cells after mechanical injury or ototoxin-induced damage [74]. Therefore, the zebrafish model has become particularly attractive for understanding molecular mechanisms underlying the development and regeneration of vertebrate sensory organs. Mutagenesis screening and functional analyses in this model have identified several RBPs potentially involved in ear development and hair cell regeneration. Our reverse genetic study has demonstrated the requirement of Rbm24a for hair cell morphogenesis in the inner ear and neuromasts [17]. IGFBP3 (insulin-like growth factor binding protein 3) is expressed in the otic vesicle at early stages of development. Using the morpholino-mediated knockdown approach, it has been shown that IGFBP3 is required for inner ear growth, hair cell differentiation, and semicircular canal formation [36]. In the developing mouse cochlea, the expression of IGFBP3 is restricted to the prosensory domain, suggesting that it may have conserved role in inner ear development [75]. However, the post-transcriptional regulatory function of IGFBP3 needs further investigation.

The zebrafish lateral line neuromast consists of a rosette of hair cells and supporting cells. Due to their accessibility, simple structure, and regenerative ability, neuromasts have been mostly used to study hair cell development and regeneration [76]. Guided mutagenesis screening has allowed the identification of several RBPs-encoding genes that are required for hair cell regeneration following ototoxin-induced damage [37]. These include *Rnpc3* (RNA-binding region containing protein 3), *Smn1* (survival motor neuron 1), and *Gemin5* (gem-associated protein 5). Rnpc3 is one of the seven integral proteins that are unique to the U12-dependent minor spliceosome, and its mutation has been associated with the human disease of isolated growth hormone deficiency [77]. Loss of Rnpc3 in zebrafish leads to pleiotropic phenotypes, mostly affecting highly proliferative tissues [78]. This suggests that Rnpc3 may regulate proliferation-related alternative splicing events during neuromast regeneration. Smn1 and Gemin5 are parts of the survival motor neuron (SMN) protein complex, which is involved in the cytoplasmic assembly of small nuclear ribonucleoproteins and in the post-transcriptional regulation of gene expression [79]. Further analysis indicates that Gemin3, another component of the SMN complex, is required for hair cell regeneration as well. Mechanistically, Smn1, Gemin3, and Gemin5 function together to inhibit the ErbB pathway, thereby inducing cell proliferation to promote regeneration of neuromasts and other tissues such as liver and tail fin [38]. These observations suggest that RBP-mediated alternative splicing events may function to promote hair cell proliferation. In light of the importance of hair cell regeneration for recovery of auditory function, it is of interest to determine whether the activity of these RBPs is conserved in other vertebrates. Interestingly, Rnpc3, Smn1, and Gemin5 are expressed in the inner ear of embryonic or postnatal mice according to the transcriptomic data available in the gEAR database [80]. Thus, it is worth identifying the targets of these splicing factors for better understanding of the post-transcriptional mechanisms by which they regulate hair cell regeneration.

SF3B4 (splicing factor 3B subunit 4) is a component of the U2-dependent major spliceosome, which plays a dominant role in removing introns from pre-mRNAs. Mutations in the *SF3B4* gene cause various diseases, including Nager syndrome (OMIM#154400), a rare congenital disorder characterized by craniofacial and limb defects [81]. Interestingly, approximately 45% of patients with *SF3B4* mutations show hearing loss [39]. Functional analyses in *Xenopus* indicate that knockdown of *SF3B4* impairs otic development by affecting gene expression in the otic vesicle [39]. Because Sf3B4 functions in both pre-mRNA splicing and in gene transcription and mRNA translation [82], how it regulates gene expression in the otic vesicle merits further investigation.

## 3. Discussion

Post-transcriptional regulation mediated by RBPs in hair cell development and regeneration is becoming increasingly important and gaining growing interest. Several RBPs are targets of key transcription factors that play critical roles during inner ear development; thus, they function in key regulatory networks to coordinate hair cell specification, proliferation, differentiation, and function. Importantly, mutations that prevent the function of several RBPs have been associated with hearing loss in humans or animal models, illustrating critical roles of these proteins in inner ear development and disease. Nevertheless, despite significant advances in this field, there are many intriguing questions that remain open for further investigation.

The mechanisms by which most RBPs function in the inner ear-specific developmental program are not entirely clear. In particular, target RNAs regulated by RBPs during inner ear development and hair cell regeneration remain elusive. While there is evidence that RBPs regulate mRNA stability and pre-mRNA splicing in the inner ear, whether they function in other post-transcriptional regulatory mechanisms, such as mRNA polyadenylation and translation, requires further investigation. Identification and functional analyses of RBP target genes could provide further insights into the molecular control of auditory and vestibular organ development. It is worth mentioning that miRNAs are post-transcriptional regulators of gene expression as well. Increasing evidence suggests that they play important roles in inner ear development and hearing loss [83]. Thus, it is important to examine the biochemical and functional interactions of RBPs with miRNAs in hair cell morphogenesis and regeneration. This should help decipher the post-transcriptional regulatory networks implicated in normal development and in diseased processes, and contribute to defining new therapeutic strategies in promoting the regeneration of hair cells and the recovery of hearing loss.

Similarly, identification of RBPs involved in regulating inner ear-specific alternative splicing events is a hotspot in understanding the post-transcriptional control of hair cell differentiation and function. Importantly, ESRP1 mutations that lead to defective alternative splicing of its target RNAs has been linked to SNHL [12]. Mounting evidence indicates that mutations in core components of the major and minor spliceosomes are responsible for spliceosomopathies representing a group of syndromes, including Nager syndrome (OMIM#154400) and thrombocytopenia-absent radius syndrome (OMIM#154400), which are caused by mutations of *SF3B4* and *RBM8A*, respectively. Interestingly, certain patients with these syndromes have SNHL [39,81], illustrating the importance of tissue-specific alternative splicing in development and disease. Clearly, inclusion or skipping of alternatively spliced exons in a large number of transcripts is critical for proper organization and functioning of stereocilia. However, how alternative splicing of most key genes essential for hearing function is regulated by RBPs remains unclear. RBP-mediated alternative splicing plays a key role in regulating the tissue-specific expression of protein isoforms. Recent studies indicate a unique splicing program along the tonotopic axis of chick cochlea, which may be regulated by splicing factors PTBP3, Esrp1, and Esrp2 [70]. It is evident that inner ear-specific alternative splicing is indispensable for hair cell development, function, and survival. For example, alternatively spliced transcripts of several members of the myosin superfamily generate protein isoforms that display distinct expression patterns and functions in the auditory system, while pathologic mutations that affect proper splicing have been associated with human nonsyndromic deafness [84]. However, despite the important interplay between alternative splicing, inner ear development, and hair cell function, the contribution of RBPs to the inner ear-specific gene expression program remains largely underestimated at present. Further understanding their functions should provide insights into the post-transcriptional regulatory mechanisms underlying hair cell development and the etiology of hearing disabilities.

Hair cell regeneration represents a promising therapeutic strategy to restore hearing. Hair cells in non-mammalian vertebrates can be regenerated from supporting cells, while hair cells in mammalian neonatal cochlea only show a limited regenerative ability [85,86]. Thus, there is a need to identify RBPs that can promote mitotic and non-mitotic hair cell regeneration. Zebrafish lateral line neuromasts represent an attractive model for mutagenesis screening and functional analysis of genes related to hair cell regeneration. Because many RBPs regulate cell proliferation or differentiation through conserved post-transcriptional mechanisms, it is worthwhile to understand whether they can function to promote regeneration of mammalian hair cells as well. Moreover, advances in genome editing technologies should allow further systematic identification of candidate genes with functions in hair cell regeneration.

## 4. Conclusions

It is becoming increasingly evident that post-transcriptional regulation of gene expression plays a critical role in inner ear development and hair cell function. Reduced mRNA stability and defective alternative splicing have been closely linked to hearing disabilities. Significant advances have been made in characterizing RBPs that are physiologically important for hair cells. However, in light of the importance of post-transcriptional events that occur in both normal development and in diseased conditions, our understanding of RBP-regulated gene expression in the inner ear remains largely elusive. Another important aspect concerns hair cell regeneration to restore noise/damage-induced hearing loss. It is of interest to identify RBPs that can promote hair cell regeneration in non-mammalian vertebrates and have conserved functions in mammals.

## Figures and Tables

**Figure 1 ijms-23-12393-f001:**
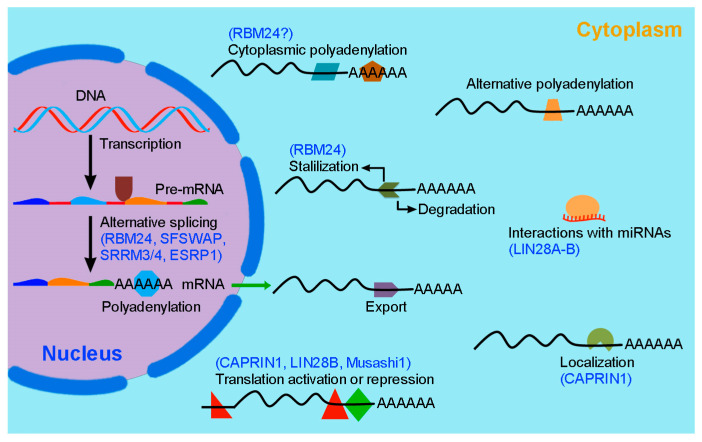
Illustration of RBP-mediated post-transcriptional regulation of gene expression. Following transcription, RBPs (colored forms) interact with pre-mRNAs, mRNAs, or miRNAs to regulate alternative splicing, polyadenylation, export, localization, stability/degradation, and translation. Shown are several RBPs with known post-transcriptional regulatory functions during inner ear development and regeneration. Note that LIN28A-B can inhibit the biogenesis of *let7* miRNA and that CAPRIN1 can sequester mRNAs into stress granules to prevent their translation.

**Figure 2 ijms-23-12393-f002:**
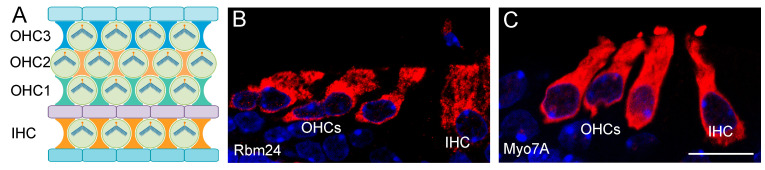
Expression of Rbm24 protein in cochlear hair cells. (**A**) Schema shows the planar cell polarity-dependent organization of hair cells in the organ of Corti [47]. One row of inner hair cells (IHC) and three rows of outer hair cells (OHC1-3) display uniform orientation of arrow-shaped stereocilia, and are rigorously aligned with surrounding supporting cells (represented by different forms and colors). (**B**,**C**) Immunofluorescence staining of cochlear sections from neonatal mice shows accumulation of Rbm24 (red) and Myo7A (red) in the cytoplasm of IHC and OHCs [44]. Rbm24-positive punctae are present in DAPI-stained (blue) nuclei. Scale bar: 10 µm.

**Figure 3 ijms-23-12393-f003:**
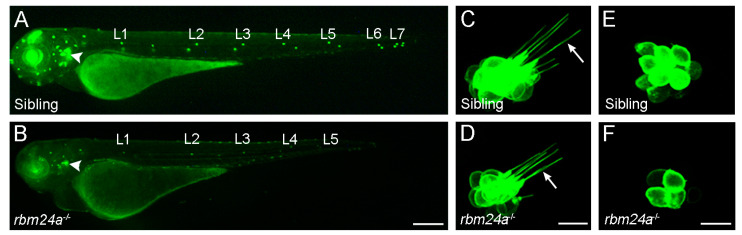
Defective hair cell morphogenesis in zebrafish *rbm24a* mutants. Hair cells in the otic vesicle and lateral line neuromasts are labeled through expression of the Tg(*pou4f3:GAP-GFP*) transgene. Adapated with permission from ref. [17]; published by Elsevier, 2020. Stereo-fluorescence images were acquired at 3 dpf (days post-fertilization) for whole embryos and hair cells. (**A**,**B**) Formation of the otic vesicle (arrowheads) and neuromasts (L1–L7) in wild-type embryos and *rbm24a* mutants. Scale bar: 250 µm. (**C**,**D**) Loss of Rbm24a impairs hair cell morphogenesis, resulting in reduced cristae size and short hair cell kinocilia (arrows). Scale bar: 10 µm. (**E**,**F**) High magnification shows defective development and organization of L1 neuromast hair cells in *rbm24a* mutants. Scale bar: 10 µm.

**Table 1 ijms-23-12393-t001:** RBPs in inner ear hair cell development, regeneration, and function.

RBPs	Functions	References
LIN28A	Upregulated after severe injury-induced progenitor regeneration in zebrafish neuromasts; required for recovery of progenitor cells and initiation of regeneration	[13]
LIN28B	Involved in hair cell production; antagonizes *let-7* miRNA and increases Akt-mTORC1 activity; required for hair cell regeneration through mitotic and non-mitotic mechanisms	[14,15,16]
RBM24	Possible transcriptional target of Atoh1 in mice; maintains the stability of *smpx*, *gsdmeb*, and *otofa* mRNAs in zebrafish; promotes inner ear-specific inclusion of exon 68 in *Cdh23* mRNA; required for hair cell morphogenesis and survival	[17,18,19,20,21]
SFSWAP	Genetic interaction with *Jagged1* in inner ear patterning; required for hearing and balance	[22,23]
QKI	Early target of noise-induced hearing loss; required for myelination of spiral ganglion neurons and auditory nerves	[24]
CAPRIN1	Transcriptional target of Pou4f3; formation of stress granules in cochlear hair cells following ototoxin-induced damage; required for synapse formation between inner hair cells and spiral ganglion neurons as well as for maintenance of the auditory function	[25,26,27]
SRRM3, SRRM4	Post-transcriptional inactivation of Rest protein in the inner ear by regulating the inclusion of exon 4 in *Rest* mRNA; required for the formation of cochlear and vestibular hair cells	[28,29,30]
ESRP1	Regulation of alternative splicing; mutations are linked to SNHL in humans (OMIM#618013); required for *Fgfr2* alternative splicing during inner ear morphogenesis and cochlear hair cell differentiation in mice	[12]
Musashi	Possible function in maintaining stem cell fate and in asymmetric division during hair cell regeneration	[31,32,33,34,35]
IGFBP3	Required for inner ear growth, hair cell differentiation, and semicircular canal formation in zebrafish	[36]
RNPC3	A component of the U12-dependent minor spliceosome; required for neuromast hair cell regeneration in zebrafish	[37]
SMN1, GEMIN3, GEMIN5	Components of the SMN complex, which induces cell proliferation by inhibiting the ErbB pathway; required for hair cell regeneration following ototoxin-induced damage in zebrafish	[37,38]
SF3B4	A component of the U2-dependent major spliceosome; mutations linked to hearing loss in humans; required for otic gene expression and otic development in *Xenopus*	[39]

## Data Availability

Not applicable.

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
