# Peer review of "Emerging Roles of RNA-Binding Proteins in Inner Ear Hair Cell Development and Regeneration"

_ijms, 2022, doi:10.3390/ijms232012393_

Round 1

Reviewer 1 Report

This review by Shi et al throws light on RNA-binding proteins and their role in inner ear development and regeneration. Traditionally, RBPs have been known to have significant implications towards post-translational modification of RNA by several processes, hence influencing gene expression. Studies on gene expression during human inner ear development are severely limited due to the complexities of its morphology. The authors have done a good job of reviewing literature to elucidate roles mainly of six RBPs – LIN28A, LIN28B, SFSWAP, QKI, ESRP1, CAPRIN1 and SRRM4. I have the following minor suggestions to this manuscript:

1.     Formatting and syntax:

a.     In Line 67, I would suggest adding LIN28 as “LIN28A-B”

b.     Please recheck font size of the text in Table 1- it appears to be larger

c.     Please include OMIM numbers for all syndromes lister

d.     Funding: Please make sure the open quotes preceding ‘The APC is appropriate.

Author Response

This review by Shi et al throws light on RNA-binding proteins and their role in inner ear development and regeneration. Traditionally, RBPs have been known to have significant implications towards post-translational modification of RNA by several processes, hence influencing gene expression. Studies on gene expression during human inner ear development are severely limited due to the complexities of its morphology. The authors have done a good job of reviewing literature to elucidate roles mainly of six RBPs – LIN28A, LIN28B, SFSWAP, QKI, ESRP1, CAPRIN1 and SRRM4.

Thanks for the positive assessment of this work.

I have the following minor suggestions to this manuscript:

Formatting and syntax:

1. In Line 67, I would suggest adding LIN28 as “LIN28A-B”

“LIN28” was changed to “LIN28A-B”.

2. Please recheck font size of the text in Table 1- it appears to be larger

The font size in Table 1 was reduced.

3. Please include OMIM numbers for all syndromes listed

Thanks for this suggestion. OMIM numbers for all syndromes cited in the article were included in the revised manuscript.

4. Funding: Please make sure the open quotes preceding ‘The APC is appropriate.

It is deleted.

Reviewer 2 Report

This is a well-written review article that describes the influence of RNA-binding proteins on development and regeneration of hair cells. It is very detailed, and will be a good reference article. Here are a few notes:

Why are RBP's unique? What is their general function? In their post-translational function, how do they compete with miRNA's? Do they interact with miRNA's at all? Do RBP's bind to microRNA's?

line 58-60. RNA-binding proteins can operate at different levels. A picture/diagram of the places where they intervene within the context of development and regeneration would be very helpful, with emphasis on the proteins that are described in this article.

line93-a diagram of Lin28B and its antagonist let7 mRNA, and interaction with Akt-mTORC1 within hair cell regeneration would help.

line 126 - what is RRM?

line 198-200 - "worth to understand" - please correct grammar

Author Response

This is a well-written review article that describes the influence of RNA-binding proteins on development and regeneration of hair cells. It is very detailed, and will be a good reference article.

Thanks for your positive assessment of this work.

Here are a few notes:

Why are RBP's unique? What is their general function? In their post-translational function, how do they compete with miRNA's? Do they interact with miRNA's at all? Do RBP's bind to microRNA's?

We addressed these questions by modifying the second paragraph of the Introduction section as follow:

“RNA-binding proteins (RBPs) are involved in the post-transcriptional regulation of gene expression and function. Through interactions with their protein partners and RNA targets including both coding and non-coding RNAs, they control RNA metabolism at multiple levels, from pre-mRNA splicing to mRNA transport, localization, stability or degradation, polyadenylation and translation”.

line 58-60. RNA-binding proteins can operate at different levels. A picture/diagram of the places where they intervene within the context of development and regeneration would be very helpful, with emphasis on the proteins that are described in this article.

Thanks for the helpful suggestion. A schema (Figure 1) illustrating the implication of RNA-binding proteins in the post-transcriptional regulation of gene expression, with emphasis on the proteins described in this article, was included in the revised manuscript.

line93-a diagram of Lin28B and its antagonist let7 mRNA, and interaction with Akt-mTORC1 within hair cell regeneration would help.

Thanks for this suggestion. The mechanism by which Lin28B/let7 axis regulates Akt-mTORC1 in cochlear epithelial cells is not clear. Therefore, we further discussed the possible role of Lin28B and let7 in mTOR signaling by citing a relevant article:

“The precise mechanism by which Lin28B and let7 regulate mTORC1 activity in cochlear epithelial cells awaits further investigation. It is possible that Lin28B promotes mRNA translation of mTOR pathway genes or relieves let7-mediated repression of their translation (Zhu et al., 2011)”.

Zhu, H.; Shyh-Chang, N.; Segrè, A.V.; Shinoda, G.; Shah, S.P.; Einhorn, W.S.; Takeuchi, A.; Engreitz, J.M.; Hagan, J.P.; Kharas, M.G.; et al. The Lin28/let-7 axis regulates glucose metabolism. Cell 2011, 147, 81–94. doi: 10.1016/j.cell.2011.08.033.

line 126 - what is RRM?

The full name “RNA recognition motif” was included in the revised manuscript.

line 198-200 - "worth to understand" - please correct grammar

It is changed to “It is also worth understanding”.

Reviewer 3 Report

This is a well-articulated review article where the authors have summarized different RNA binding proteins and their role in hair cell development and regeneration.  I would like to congratulate the authors for their effort in putting together a fantastic summary and review of most of the RBPs known in this context. Following are few minor comments to be addressed by the authors.

1.     The authors might consider adding expression of Musashi1and its role in hair cell regeneration.

2.     In section 2.7 the authors have combined an exhaustive list of RBPs that show that have potential role in hair cell regeneration. From the transcriptomic data available in the gEAR database, it suggests that some of these (like Rnpc3, Smn1 and Gemin5) are also expressed in either embryonic or postnatal mouse cochlea and vestibular epithelia in hair cells or supporting cells. The authors might include that in their description.

3.     In the same section 2.7, for IGFBP3, mention that it is also expressed in the developing otic vesicle and add the reference of Okano and Kelley 2013.  

Author Response

This is a well-articulated review article where the authors have summarized different RNA binding proteins and their role in hair cell development and regeneration. I would like to congratulate the authors for their effort in putting together a fantastic summary and review of most of the RBPs known in this context.

Thank you for the positive assessment and for your enthusiasm.

Following are few minor comments to be addressed by the authors.

1. The authors might consider adding expression of Musashi1 and its role in hair cell regeneration.

Thanks for raising this important point that helps to improve the manuscript, and we apologize for missing Musashi1 in our previous discussion. A new section discussing its function in hair cell development and regeneration was included in the revised version, with the following references:

1). Sakaguchi, H.; Yaoi, T.; Suzuki, T.; Okano, H.; Hisa, Y.; Fushiki, S. Spatiotemporal patterns of Musashi1 expression during inner ear development. Neuroreport 2004, 15, 997–1001. doi: 10.1097/00001756-200404290-00013.

2). Murata, J.; Murayama, A.; Horii, A.; Doi, K.; Harada, T.; Okano, H.; Kubo, T. Expression of Musashi1, a neural RNA-binding protein, in the cochlea of young adult mice. Neurosci. Lett. 2004, 354, 201–204. doi: 10.1016/j.neulet.2003.10.036.

3). Savary, E.; Hugnot, J.P.; Chassigneux, Y.; Travo, C.; Duperray, C.; Van De Water, T.; Zine, A. Distinct population of hair cell progenitors can be isolated from the postnatal mouse cochlea using side population analysis. Stem Cells 2007, 25, 332–339. doi: 10.1634/stemcells.2006-0303.

4). Wakasaki, T.; Niiro, H.; Jabbarzadeh-Tabrizi, S.; Ohashi, M.; Kimitsuki, T.; Nakagawa, T.; Komune, S.; Akashi, K. Musashi-1 is the candidate of the regulator of hair cell progenitors during inner ear regeneration. BMC Neurosci. 2017, 18, 64. doi: 10.1186/s12868-017-0382-z.

5). Kinoshita, M.; Fujimoto, C.; Iwasaki, S.; Kashio, A.; Kikkawa, Y.S.; Kondo, K.; Okano, H.; Yamasoba, T. Alteration of Musashi1 intra-cellular distribution during regeneration following gentamicin-induced hair cell loss in the guinea pig crista ampullaris. Front. Cell. Neurosci. 2019, 13, 481. doi: 10.3389/fncel.2019.00481.

6). Okano, H.; Kawahara, H.; Toriya, M.; Nakao, K.; Shibata, S.; Imai, T. Function of RNA-binding protein Musashi-1 in stem cells. Exp. Cell Res. 2005, 306, 349–356. doi: 10.1016/j.yexcr.2005.02.021.

2. In section 2.7 the authors have combined an exhaustive list of RBPs that show that have potential role in hair cell regeneration. From the transcriptomic data available in the gEAR database, it suggests that some of these (like Rnpc3, Smn1 and Gemin5) are also expressed in either embryonic or postnatal mouse cochlea and vestibular epithelia in hair cells or supporting cells. The authors might include that in their description.

Thanks for this suggestion. We added the following sentence:

“Interestingly, Rnpc3, Smn1 and Gemin5 are also expressed in the inner ear of embryonic or postnatal mice according to the transcriptomic data available in the gEAR database (Orvis et al., 2021)”.

Orvis, J.; Gottfried, B.; Kancherla, J.; Adkins, R.S.; Song, Y.; Dror, A.A.; Olley, D.; Rose, K.; Chrysostomou, E.; Kelly, M.C.; et al. gEAR: Gene expression analysis resource portal for community-driven, multi-omic data exploration. Nat. Methods 2021, 18, 843–844. doi: 10.1038/s41592-021-01200-9.

3. In the same section 2.7, for IGFBP3, mention that it is also expressed in the developing otic vesicle and add the reference of Okano and Kelley 2013.

We added the following sentence:

“In the developing mouse cochlea, the expression of IGFBP3 is restricted to the prosensory domain, suggesting that it may have conserved role in inner ear development (Okano and Kelly, 2013)”.

Okano, T.; Kelley, M.W. Expression of insulin-like growth factor binding proteins during mouse cochlear development. Dev. Dyn. 2013, 242, 1210–1221. doi: 10.1002/dvdy.24005.